# Gender, caste, and heterogeneous farmer preferences for wheat varietal traits in rural India

**Vijesh V. Krishna**[1]*, **Prakashan C. Veettil**[2]

**1** International Maize and Wheat Improvement Center (CIMMYT), Hyderabad, India, **2** International Rice Research Institute (IRRI), New Delhi, India

* v.krishna@cgiar.org

## Abstract

The research on crop genetic enhancement has created a continuous flow of new, improved germplasm for the benefit of farmers and consumers of the Global South during and after the Green Revolution. Understanding farmers' heterogeneous preferences for varietal traits in different market segments and incorporating the prominent ones in crop breeding programs are expected to facilitate a faster diffusion of these new varieties. Albeit knowing little about farmers' trait preferences in South Asia, public-sector breeding programs prioritize yield enhancement and risk reduction over other varietal traits. Against this backdrop, we examined wheat farmers' preferences for varietal traits in Central India, where the prevailing varietal turnover rate has been meager. We conducted a ranking exercise among 120 individuals, followed by a sex-disaggregated survey with a choice experiment among 420 farmhouseholds in 2019. The lowest varietal turnover rate was observed for the socially marginalized castes. Most women respondents were not actively involved in making decisions related to wheat cultivation, including varietal selection. However, the results indicate that marginalized caste and women farmers are open to experimentation with new varieties, as shown by their positive willingness to pay for improved varietal traits. Across the gender and caste groups, grain quality attributes (especially *chapati* quality) were ranked high, above the yield-enhancing and risk-ameliorating traits. From the observed patterns, one could deduce that developing and disseminating improved varieties with better grain quality and targeting women and marginalized social groups in varietal dissemination programs could enhance farmer adoption of new, improved germplasm and wheat productivity in Central India.

## 1. Introduction

Germplasm improvement is an important mechanism for increasing the potential yields [1, 2], averting the yield losses from biotic and abiotic stresses [3, 4], and enhancing the consumption and nutritional value of crops [5, 6]. Nevertheless, all improved crop varieties are not equally accepted by farmers. Within a seed dissemination system, the likelihood that a new variety

**Data Availability Statement:** All relevant data are within the paper and its Supporting Information files.

**Funding:** This work was supported by the Indian Council of Agricultural Research (ICAR) in

collaboration with CIMMYT-India, and the CGIAR Research Program on Wheat Agri-food Systems (WHEAT; https://wheat.org/). The donor agencies had no role in study design, data collection and analysis, decision to publish, or preparation of the manuscript.

**Competing interests:** The authors have declared that no competing interests exist.

would outperform farmers' expectations with respect to different key traits determines its adoption [7]. Yield enhancement and sustenance have remained the most crucial breeding goals for most crops [8, 9], whereas socioeconomic research suggests that farmers often base their adoption decisions on non-yield traits [10]. Furthermore, varietal traits may be valued differently by different farm households and even by different household members [11–13]. For crops like wheat, we know little about such preference heterogeneities and their implications [14]. Recognizing the importance of closing this knowledge gap, we undertook this study, eliciting farmers' preferences for a set of wheat varietal traits. We investigated the presence of intra-household (gender-based) and inter-household (caste-based) preference heterogeneities among wheat farmers. We analyzed the primary data collected from Central India, one of the largest wheat-producing regions of South Asia, but with wheat productivity lying below the national average.

Defining the gender-based trait preferences, determining the heritability of traits, and assessing the genetic, economic, and cultural trade-offs between different traits form the basis of a gender-responsive plant breeding program [15]. The existence of gendered preferences has been widely examined in the literature for several crops, such as rice [12], maize [16], banana [13], and cassava [17]. However, not many have examined the nature of gendered preferences for wheat varietal attributes. Similarly, the differential demand for technologies across caste groups and the caste system's role in varietal change have hardly been discussed for any crop. The few exceptions are Arora et al. [18] and Khatri-Chhetri et al. [19], which indicated that different caste groups valued new, improved varieties differently. Non-marginalized groups were generally found to value most varietal attributes higher than marginalized groups. However, the effect of caste may differ between regions, crop types, and traits against which the preferences are elicited.

Understanding the preference heterogeneity within the farming community for various varietal attributes is essential for ensuring farmers' participation in decision-making at all stages of plant breeding, which has been acknowledged as a critical strategy for promoting and upscaling new varieties [20]. Together with substantial economies of scale, public-sector wheat breeding tends to be centralized (thereby relatively non-participatory) in several countries [21]. At the same time, a sizeable wheat area in the Global South is still cultivated with old varieties [22–24]. Although varietal change has accelerated recently in several parts of the developing world because of farmer demand for new resistant varieties to manage the rust outbreaks [25], wheat varieties released in the early 1980s are still prevalent in less rust-affected areas like Central India [26].

An empirical evaluation of farmer preference for wheat varieties is not only relevant for the topic being less explored in the literature. The socioeconomic landscape of Central India provides a unique context to assess the heterogeneity in farmer preferences toward technology acceptance due to the social and economic marginalization. We will see in the forthcoming sections that a large share of the wheat production by sample households–especially among the income-poor–goes for home consumption, making consumption utility an integral part of the decision-making over varieties. The popularity of some of the old wheat varieties (e.g., Lok-1, released in 1982) in the region has been attributed to their suitability for making tasty *chapatis* (and *chapati* variants such as *roti*, *paratha*, *phulka*, etc.) [26, 27]. However, no study has established the connection between farmers' preference for consumption traits and the prevalence of old varieties, particularly among the income poor and food insecure, the socially marginalized, and women farmers.

The paper is structured as follows. The next section includes a literature review on preference heterogeneity for varietal traits in agriculture, from which testable hypotheses are developed. The methodological framework, including a description of the study area, sampling

strategy, and econometric estimation, is presented in Section 3. The results of the empirical data analyses are presented in Section 4 and discussed in Section 5. The last section concludes the findings.

## 2. Context and hypotheses

The core theme of this study is preference heterogeneity, i.e., the extent to which individual tastes and preferences vary within a society. The determinants and implications of heterogeneities among individuals are examined in the applied economics literature on various topics. In the discipline of agricultural economics and rural development, farmer preferences are frequently elicited to explain the relative importance of various technology traits. A quick literature review shows that preference heterogeneity has been examined for several crops with respect to varietal adoption. Maligalig et al. [12] examined the heterogeneity in farmer preferences for improvements in rice variety traits, using data gathered from experimental investment games conducted in the Philippines. Marimo et al. [13] conducted a systematic review of banana trait preferences. Acheampong et al. [28] studied Ghanaian cassava farmers' selections of varieties and found dominant preferences for the longevity of root storage in soil and resistance to disease. Using participatory rural appraisal among Ethiopian sorghum farmers, Derese et al. [29] highlighted farmers' preference for medium-maturing varieties with high grain and biomass to escape post-flowering drought. Teeken et al. [17] observed significant preference heterogeneity for cassava varietal traits in Nigeria: women preferred varieties with superior cooking traits and men varieties with better agronomic traits. Kassie et al. [30] estimated taste parameters and heterogeneities among maize farmers of Zimbabwe and observed drought tolerance, grain yield, and cob size as among the most favored traits.

Not many studies have been conducted on farmer demand for wheat varietal traits. In general, the literature on the socioeconomic analysis of wheat production and marketing has remained relatively thin compared to other cereal crops [31]. Most preference elicitations were conducted in Ethiopia [32–36], while information on wheat farmers' preferences from extensive and diverse wheat-producing tracts of India is limited. Although the empirical studies from neighboring countries–including Ortiz-Ferrara et al. [37] and Bhatt et al. [38] in Nepal–provide valuable insights into the attributes preferred by South Asian farmers, they are few in number and lack a focus on preference heterogeneity. The differences in the sociopolitical landscape of India from Nepal could affect both the speed of varietal turnover and hence demand a country-specific evaluation of farmer demand for varietal traits.

In the present study, we analyze the relative importance of grain quality, yield (quantity), and production risk reduction as perceived by farmers in Central India, where varietal turnover is sub-optimal (21 years [26]). Some of the existing studies on varietal release and adoption suggest that the superior consumption quality of old varieties (such as Lok-1) could be one of the main reasons for farmers' consistent demand for them [23]. Panghal et al. [39] assessed the quality of popular wheat varieties and found that Lok-1 has a high thousand-kernel weight, hectoliter weight, and better softness and puffing height (traits associated with superior *chapati* quality). Another popular variety in the Central Zone, HI-617 (*Sujata*), was also described as superior in this regard [27]. Based on these observations, we generated the following hypothesis:

- 'Smallholder wheat farmers in rural areas value grain quality (e.g., suitability for *chapati*-making) equal to or above yield enhancement and other varietal attributes.'

Our second hypothesis deals with gender- and caste-based preference heterogeneity. For non-wheat crops, several studies conducted outside India have shown preference

heterogeneity between male and female farmers due to the difference in the intended end-use of the crop, such as profit generation *versus* subsistence consumption [29, 40–42]. There is a consensus in the literature that men often prefer 'economic traits' such as high yield so that the surplus production can be sold, whereas women's reproductive role influences them to prioritize food security and household taste preferences. However, the manifestation of these preferences in varietal choices varies across agrarian societies, as it depends on women's role in decision-making. Some researchers point to a scenario shift in South Asian agriculture with women increasingly taking part in the management and decision-making in wheat production [43]. We aim to explore intra-household heterogeneity in preferences for wheat varietal traits based on the gender of the respondent and describe how this could be manifested in the form of varietal selection by the wheat farmers of Madhya Pradesh. The testable hypothesis made against this backdrop is given below:

- 'There are intra-household gendered differences in preferences for wheat varietal traits. Within a household, traits preferred by men are different from those preferred by women.'

We proceed further by examining inter-household heterogeneity in preferences, focusing on the social institution of caste. The caste system is hereditary, exhaustive, and omnipresent in India's urban and rural social lives, clustering society into several (mostly) endogamous groups [44, 45]. Caste-based social segregation is a unique feature of South Asian communities, determining the social norms and affecting millions of livelihoods in rural and urban India [46, 47]. There is ample evidence that caste determines farmers' potential to generate income through several pathways, including technology adoption, information access, and social learning [48–50]. However, caste-based heterogeneity of demand for technologies has not often been examined, with a few exceptions, such as the studies by Khatri-Chhetri et al. [19] and Arora et al. [18]. Here, we examine whether varietal age is higher for marginalized castes and whether preference heterogeneity exists with respect to caste by testing the following hypothesis:

- 'Significant inter-household preference heterogeneity exists for wheat varietal traits across caste groups.'

These three hypotheses are tested using primary data collected by interviewing 818 members of 420 wheat-farming households in the Madhya Pradesh state of India during July-September 2019.

## 3. Analytical framework

### 3.1. Study area

The primary data were collected from the central Indian state of Madhya Pradesh, the second-largest state of India by area and the fifth by population. The state's agriculture is characterized by a highly diverse cropping system, which is the primary source of livelihood for the rural population [51]. Madhya Pradesh accounts for about one-fifth of the national wheat area, despite experiencing a low productivity level at 2.8 tons/ha, compared with the states of the western Indo-Gangetic Plains (IGP), like Punjab with 4.3 tons/ha and Haryana with 4.0 tons/ha [52]. The surveys were conducted in Jabalpur, Damoh, and Mandla districts. Despite its historical, industrial, and political relevance, the region has high poverty and homelessness and lacks infrastructure development [53]. The study area also has an interesting diversity concerning caste. According to the 2011 Census data, Madhya Pradesh has many marginalized castes and communities [54]. About 21% of the population (15 million) are scheduled tribes (ST), while 16% (11 million) belong to scheduled castes (SC). For ease of representation and analysis, we combine SC and ST categories into a single group (SCST) in this paper. There are

numerous individual castes and sub-castes within the caste categories with unique traditions, norms, and belief systems [55]. Poverty has been more prevalent among the SCST households of Madhya Pradesh as compared to the other caste groups in the state, as well as the SCST households living in most other Indian states [56].

## 3.2. Sampling and data collection

The primary data were collected during July-September 2019 as a second-round household survey among the wheat farmers who were already interviewed in the previous year by the first author. Most of the existing socioeconomic analysis of wheat cultivation in India is either from the western states of the IGP, such as Punjab and Haryana, or from the eastern states, such as Bihar. The state of Madhya Pradesh and the three districts were selected purposively, recognizing the knowledge gap on wheat cultivation in the subsistence farming systems of Central India. The selected districts present an interesting spectrum of social diversity, where the tribal and non-tribal households live in close geographical proximity [54].

The sampling procedure followed in the first-round (2018) can be elaborated as follows. From each district, we selected four villages randomly. However, three out of the twelve selected villages had small populations (i.e., 75 households or less), and it was challenging to find enough wheat farmers to include in the household surveys. To compensate, an additional three small villages were added from the same districts. In addition to these fifteen randomly selected villages, three villages were included where qualitative gender studies were completed in 2014 and 2015 as part of a project, GENNOVATE (https://gennovate.org/), which was a large-scale qualitative comparative research initiative to analyze how the nexus between gender norms and women's agency in agriculture influenced the diffusion of different innovation processes across diverse contexts [57]. An overlap of the study villages was expected to generate useful insights on the locally prevailing social norms from the existing GENNOVATE case studies. From all the villages, a census of farm households was completed in September 2018, from which wheat farming households were selected randomly for the survey. In total, 28 farm households were included from each of the nine large villages (336 households) and 14 households from each of the six small villages (84 households) in October-November 2018. These farmers were re-approached for the second round of the survey, conducted in September-October 2019. None of the GENNOVATE focus group discussants fell in the random sample of households. For details of the sampling procedure, one may refer to the papers developed from the 2018-dataset [58, 59].

Because the choice experiment on farmer preference for wheat varieties was included only in the 2019-round, we excluded the 2018 dataset from the main analysis. However, the 2018-round data was used for complementary analysis on gendered participation in wheat varietal selection. This analysis provided the necessary contextual reasoning for the heterogeneity in the trait preferences.

The fieldwork of 2019 started with a preliminary ranking exercise of varietal traits among a sub-sample of 120 (both men and women) farmers, randomly selected from the 2018 respondent list. These farmers were asked to rank the 12 key attributes of wheat varieties obtained from the literature review and expert consultations and were presented to the respondents with representative photos. Similar ranking exercises have been commonly conducted to elicit the trait preferences for different crops [60, 61]. The insights from the ranking exercise were used to construct a choice experiment, data from which form the core of our study. For the main survey, interviews were conducted separately with one male and one female member from the households covered in the 2018 household survey. A few households had members of only one sex, and if so, only one interview was conducted. The final sample in our study

consisted of 818 respondents (413 women and 405 men) from 420 households. The interviews were carried out by a team of 14 trained (8 female) enumerators and two supervisors (both male) in September 2019, using Computer-Assisted Personal Interview (CAPI) software. Informed consent was sought from each participant before the interview. On average, an interview was completed within 90 minutes.

## 3.3. Choice experiment design

One of the necessary steps in designing a choice experiment is to identify the relevant attributes (varietal traits) for each alternative and the levels of each attribute considered for the experiment [62, 63]. Discussions with local experts were carried out to understand the possible reasons for farmer preference for certain traits in the ranking exercise and to fix realistic levels of each of the selected attributes. Using the insights, five traits were included in the choice experiment:

i.  Potential grain yield under ideal circumstances (representing quantity);

ii.  Suitability for *chapati* making (quality trait);

iii.  Tolerance to terminal heat (a less-familiar risk-ameliorating trait);

iv.  Lodging tolerance (a more-familiar risk-ameliorating trait);

v.  Seed price.

The details of shortlisting the attributes for the choice experiment are provided in the result section. For each attribute, three levels were provided; these are described along with units and levels in Table 1. In order to mitigate possible bias resulting from inappropriate attribute levels in the experiment, we carried out five multi-locational focus group interviews in the study area to finalize the attribute levels. A full factorial design with these attributes and identified levels would generate a large number of choices. A D-efficient design was used to identify the optimum combination of choice sets while estimating the main effects without losing any information. An optimal, fully fractional design assuming zero priors consisting of a single block and nine choice sets was selected. Each choice set offered respondents an option to choose from four alternatives (three product profiles and the status quo), resulting in a multidimensional choice-making scenario where respondents considered all attributes simultaneously. In the piloting stage, we found that increasing the alternatives per choice set above three (plus the status quo option) significantly added interview time. Providing two alternatives and the status quo was also considered. However, we followed the suggestion from the literature that providing three alternatives could yield more robust models than two [64]. When two alternatives are provided, serial non-participation–"choosing an alternative consistently without regard for changes in the attributes" (ibid; p. 1140)–was found to occur.

The inclusion of the status quo (adoption of wheat varieties currently cultivated by the respondents) avoided a forced-choice by providing the possibility to choose none of the alternatives in the choice set, making the design consistent with the theory of demand. The cards' illustrations were pre-tested and corrected to mitigate any possible cognitive difficulties among the sampled respondents before the main survey. An example choice card is provided in S1 Appendix and the complete set in the S1 File. The choice card descriptions were translated into Hindi, the main local language.

When presented with less-familiar crop choices, the lack of information about the costs and benefits affects the respondents' choices, resulting in high attribute non-attendance (ANA). In this choice behavior, respondents overlook one or more attributes in the choice experiment questions while making decisions [65]. The ANA framework in the analysis has special

**Table 1. Attributes and levels in the choice experiment design for wheat varietal preference.**

| Attributes | Description | Levels |
|---|---|---|
| Heat tolerance | Loss of wheat yield due to terminal heat stress during milking and grain-filling stage (as the percentage of yield from non-stress years) | [1] 10% yield reduction<br>[2] 20% yield reduction<br>[3] 30% yield reduction |
| Potential yield | Potential yield achievable under ideal management conditions (quintals/acre) | [1] 8 quintals (= 2 tons/ha)<br>[2] 12 quintals (= 3 tons/ha)<br>[3] 16 quintals (= 4 tons/ha) |
| *Chapati* quality | Quality of *chapati* made from the wheat flour (ordered category) | [1] Low quality<br>[2] Medium quality<br>[3] High quality |
| Seed price | Price of wheat seed presented (Rs./acre)[#] | [1] Rs. 1000 (= US$ 35.1/ha)<br>[2] Rs. 1120 (= US$ 39.3/ha)<br>[3] Rs. 1280 (= US$ 44.9/ha) |
| Lodging tolerance | Wheat varieties' tolerance to lodging (ordered category) | [1] Low tolerance<br>[2] Medium tolerance<br>[3] High tolerance |

Notes

[#] 1 US$ = Rs. 70.42 in 2019 (https://data.worldbank.org/indicator/PA.NUS.FCRF).

relevance for the present study, as it was conducted among less-educated farmers. The low cognitive capacity of some of the respondents may lead to ignoring attributes while choosing, and the choice sometimes can be heuristic in nature. If so, the WTP estimates may suffer from hypothetical bias, a common issue in several discrete choice experiments. If respondents ignore some of the attributes presented in the choice cards, the associated welfare values would be precluded from the analysis, leading to biased estimates [66]. We have also employed information priming on terminal heat stress, a least familiar trait for farmers, with a pictorial representation based on Farooq et al. [67] (Fig 1).

## 3.4. Econometric estimation of farmers' willingness to pay

Let us start with the assumption that a farmer's utility depends on the various choices made from a set of alternatives. Employing the random utility framework, each respondent's indirect utility function can be partitioned as deterministic (observable) and random (unobservable). Hypothesizing that the utility function is systematically different for men and women (or for marginalized and non-marginalized castes), we estimated different econometric models, modifying the distributional assumption on the random part of utility. We assume that the observable part of the utility, when an alternative variety is being chosen, depends on the seed price ($P$) and other variety attributes ($Z$), along with other idiosyncratic variables such as farmers' socioeconomic characteristics. Empirically, we specify it as follows:

$$VAR_{ijc} = \alpha_{1i}(\text{Potential Yield})_{ijc} + \alpha_{2i}(\text{Chapati Quality})_{ijc} + \alpha_{3i}(\text{Lodging Tolerance})_{ijc}$$
$$+ \alpha_{4i}(\text{Heat Tolerance})_{ijc} + \delta_i(\text{Seed Price})_{ijc} + \varepsilon_{ijc} \qquad \text{(Eq 1)}$$

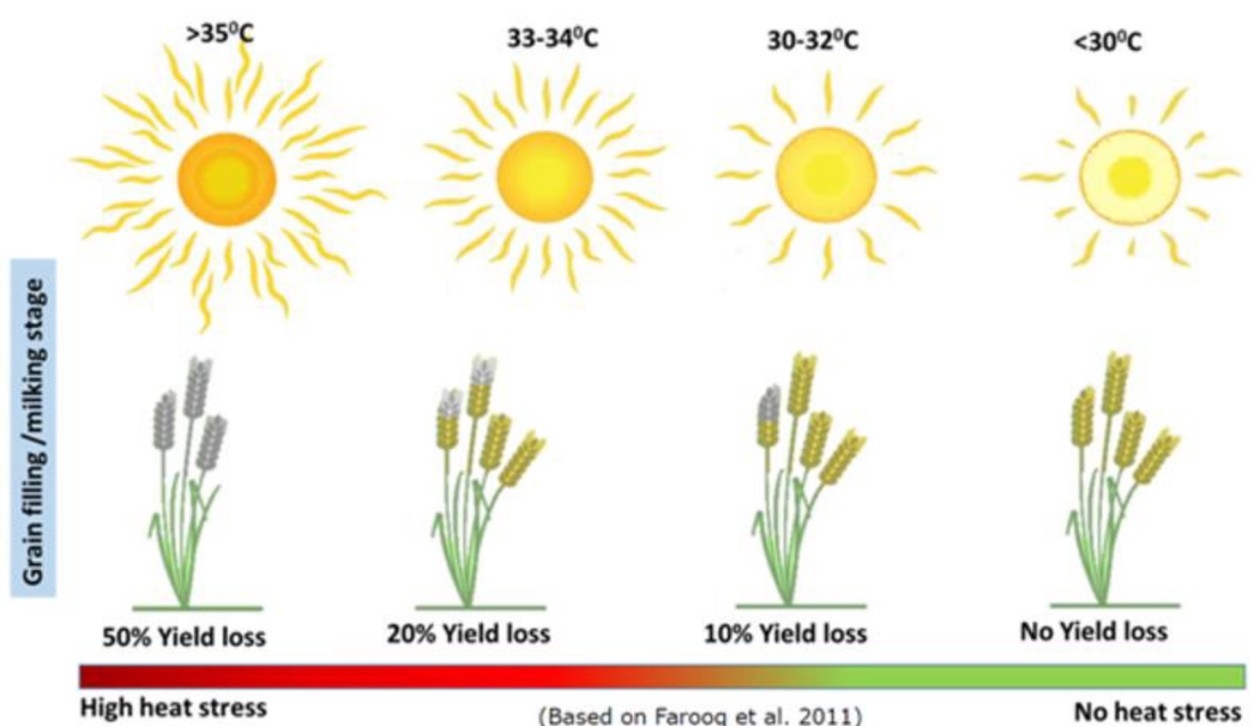

**Fig 1. Information priming on the impact of late heat stress in wheat.** *Note*: Developed by Sreejith Aravindakshan, based on Farooq et al. [67].

where $VAR_i$ denotes the discrete choice made by farmer $i$, which takes a value of 1 if farmer $i$ chooses wheat variety $j$ in the choice set $C$ (= 0 otherwise). We have to estimate $\alpha_1$, $\alpha_2$, $\alpha_3$, and $\alpha_4$, which are the parameters for design attributes: yield, *chapati* quality, lodging tolerance, and heat tolerance, respectively, and $\delta$ is the coefficient of seed price (cost of seeds per land unit).

Preference for improved wheat varieties as an alternative to the status quo and the value of the variety attributes might vary depending on a farmer's level of gender (binary variable with 1 = female), illiteracy (binary variable with 1 = respondent cannot read or write), caste category (binary variable, with 1 = SCST households), and level of food insecurity ('Food Insecurity' score, from 1–8), landholding size (ha), and age of the respondent (years). Food insecurity was estimated with Household Food Insecurity Access Scale, which is commonly used to measure household food insecurity [68, 69]. The equation included two interaction terms, gender with SCST dummy and Food Insecurity score, to capture the gender intersectionality in preference formation with farmer's caste and economic status. We incorporate these vectors of variables by interacting with the alternative specific constant (ASC) for an improved variety alternative.

$$\begin{aligned} VAR_{ijc} = {} & \alpha_{1i}(\text{Potential Yield})_{ijc} + \alpha_{2i}(\textit{Chapati}\ \text{Quality})_{ijc} + \alpha_{3i}(\text{Lodging Tolerance})_{ijc} \\ & + \alpha_{4i}(\text{Heat Tolerance})_{ijc} + \delta_i(\text{Seed Price})_{ijc} + \tau_{1i}(\text{Gender})_{ijc} + \tau_{2i}(\text{SCST})_{ijc} \\ & + \tau_{3i}(\text{Gender.SCST})_{ijc} + \tau_{4i}(\text{FoodInsecurity})_{ijc} + \tau_{5i}(\text{FoodInsecurity.Gender})_{ijc} \\ & + \tau_{6i}(\text{Age})_{ijc} + \tau_{7i}(\text{Illiteracy})_{ijc} + \tau_{8i}(\text{Landholding})_{ijc} + \varepsilon_{ijc} \qquad (\text{Eq 2}) \end{aligned}$$

In Eq (2), $\tau_1$, $\tau_2$, ... are parameters of key socioeconomic variables, which need to be estimated. We estimated Eq (1) and Eq (2) using a random parameter logit model using the maximum simulated likelihood approach [70]. We assumed that preference heterogeneity exists, and thus the attributes are specified randomly and normally distributed, with parameters $\tau_1$,

$\tau_2,\ldots$ are considered non-random. The mixed logit models were estimated with interaction terms to compare the average WTP of women with that of men. Here, the ASC is interacted with the female dummy variable. A positive ASC interaction with a household attribute would imply that if the respondent did not choose the status quo option in the choice experiment, he or she would derive a positive utility from adopting the new varieties and be willing to pay for them. The estimation was done using the R packages, mlogit [71], and gmnl [72].

Deviating from the traditional approach of calculating the willingness to pay (WTP) for an attribute and employing the extension of parameter space to the WTP space approach suggested by Train and Weeks [73] and Sonnier et al. [74], we allowed random parameters (and, consequently, random WTP measures) in the estimation. This approach, as it accounts for the preference heterogeneity [73], is more suitable for the current study over the alternative (the preference space approach). The re-parameterization-based estimation of marginal WTP in the WTP space models allows us to estimate the heterogeneity distribution directly. One may refer to Scarpa et al. [75] for details on this approach. Following Scarpa et al. [65], marginal WTP space is estimated for Eqs (1) and (2) for those farmers who did not ignore all attributes while making choices.

# 4. Results

## 4.1. Descriptive statistics

We present the popular wheat varieties cultivated in the study area in Table 2. Four varieties were cultivated by about 88% of the sample households: Lok-1 (a 36-year-old variety, associated with superior *chapati*-making qualities and wide adaptability); GW-322 (a 16-year-old variety resistant to rust diseases and suitable for *chapati*-making); HI-617 (a 36-year-old variety, popularly known as *Sujata*, also associated with superior grain quality); and GW-273 (a 20-year-old variety, resistant to rust diseases). The average age of wheat varieties, calculated as the difference between the year of sowing in the study season (i.e., 2018) and the year of the official release of the variety in the country, was 30 years for the whole sample. The estimates also showed significant inter-household differences in varietal use. The average varietal age was significantly higher among SCST farmers (34 years) than among other caste farmers (28

**Table 2. Wheat varieties grown in the study area.**

| Varietal name | Varietal age, in years | Share of sample households cultivating the variety | | |
|---|---|---|---|---|
| | | Overall [n = 408] | SCST [n = 112] | Other castes [n = 296] |
| Lok-1 | 36 | 0.439 | 0.473 | 0.426 |
| GW-322 | 16 | 0.179 | 0.054 | 0.226*** |
| HI-617 (*Sujata*) | 36 | 0.162 | 0.357 | 0.088*** |
| GW-273 | 20 | 0.103 | 0.036 | 0.128*** |
| HD-2851 (*Pusa Vishesh*) | 13 | 0.022 | 0.000 | 0.030* |
| HD-4672 (*Malva Ratna*) | 18 | 0.020 | 0.009 | 0.024 |
| C-306 | 49 | 0.017 | 0.018 | 0.017 |
| WH-147 | 40 | 0.015 | 0.009 | 0.017 |
| Other varieties | 26.33 (17.71) | 0.044 | 0.045 | 0.044 |
| Mean varietal age of all wheat varieties, in years | | 29.77 (9.88) | 34.16 (7.12) | 28.11*** (10.28) |

*Notes*: The details of the most important wheat variety cultivated on-farm were elicited. Varietal age is calculated as the difference between the reference year (here, 2018) and the year of official varietal release in the country. Figures in the parentheses show the standard deviation of the sample mean values (weighted by their sample share). Here, n stands for the number of wheat-growing households (as of 2018/19). *** and * denote statistical significance of the difference from the SCST category at the 0.01 and 0.10 levels, respectively.

years). The popularity of a relatively old variety, HI-617, was significantly high among SCST farmers (36% considered this as the major variety cultivated on-farm) than other castes (9%). On the other hand, a relatively new variety, GW-322, was cultivated by only a smaller share of SCST farmers (5%) than the others (23%).

In India, farms are managed jointly by several household members, and it is rare to observe plots managed by women and men separately, unlike in many other parts of the world. However, the male household head often dominates in the decision-making concerning food crop production. While women were active in food crop production activities (proving family labor, for example) in 97% of the sample households, only 8% had a woman in charge of decision-making (Table 3). In contrast, almost all male farmers were involved in production activities, and about 74% were responsible for decision-making also. A majority of women were excluded from discussions conducted within the household before making decisions. Being devoid of agency and excluded from these discussions, women had only limited awareness of wheat farming activities, which could be why 43% of them could not name a major wheat variety that they had recently cultivated.

We supplement Table 3 with data on gendered participation in wheat varietal selection, using the 2018 survey data (S2 Appendix). Across the caste groups, in more than 90% of households, men were responsible for varietal selection. Although decision-making involved a discussion among the household members in most cases (>84%), only 30–45% of these discussions involved women. Also, most male respondents did not acknowledge the role of women members of the household in the discussions on varietal selection. For instance, women respondents from 144 households indicated that they participate in the discussions related to varietal selection. In about 58% of these cases, the male respondents denied their involvement. Due to a lack of agency and male household heads' denial of their involvement in varietal selection, women farmers' varietal preferences might not be fully expressed in the adoption statistics, making a gendered preference elicitation an important addition to the literature and for the seed market development strategies.

**Table 3. Role of husbands and wives in wheat producing farm-households across the gender and caste groups.**

| [dummy variables; 1 = yes; 0 = otherwise] | | Whole sample [n = 740] | Male respondents (husband) | | | Female respondents (wife) | | |
|---|---|---|---|---|---|---|---|---|
| | | | Overall [n = 370] | SCST [n = 107] | Other castes [n = 263] | Overall [n = 370] | SCST [n = 107] | Other castes [n = 263] |
| Respondent is involved in food crop production activities (providing labor or involved in management) | | 0.984 | 0.995 | 0.991 | 0.996 | 0.973** | 1.000 | 0.962***,## |
| Respondent is solely responsible for making decisions related to food crop production by the farm household | | 0.411 | 0.739 | 0.792 | 0.718 | 0.075*** | 0.093*** | 0.067*** |
| Among respondents not solely responsible for the decisions related to food crop production, the extent of involvement in the decision-making process | (i) Not involved | 0.064 | 0.009 | 0.000 | 0.013 | 0.079*** | 0.025 | 0.106**,### |
| | (ii) Involved in some of the decisions | 0.600 | 0.396 | 0.323 | 0.427 | 0.658*** | 0.667*** | 0.654*** |
| | (iii) Involved in most or all decisions | 0.337 | 0.594 | 0.677 | 0.560 | 0.262*** | 0.308*** | 0.240*** |
| Respondent knew the name of the major wheat variety cultivated on their farm | | 0.766 | 0.967 | 0.972 | 0.966 | 0.565*** | 0.561*** | 0.567*** |

*Notes*: The inter-group equality of proportions is tested using large-sample statistics.

** and *** denote the statistical significance of difference from the male group at the 0.05 and 0.01 levels, respectively.

## and ### denote the statistical significance of difference from the SCST group (within the gender grouping) at the 0.05 and 0.01 levels, respectively.

## 4.2. Farmer preference for varietal traits

As the first step in preference elicitation, a subsample of respondents (n = 120) was asked to rank those attributes that they had indicated as 'relevant' in selecting wheat varieties for their farm. About 58% of the respondents were women, 29% belonged to one of the marginalized castes (SC or ST), and 42% owned less than 1 ha of cultivable landholding. The findings from the ranking analysis, both the share of farmers who cited a trait as essential and the rank they provided for each trait, are shown in Table 4.

According to the ranking exercise, the two most important characteristics of wheat varieties were related to quality–good for *chapatis* (taste, puffiness, etc.) and color of grains at maturity (light-colored, white, or yellow). About 94% of respondents found these traits important in varietal selection. Big/bold grain, a composite trait of quality and quantity, was important for 86%. Length and spread of awns–another composite trait of quality and quantity (awn spread indicates boldness of grains) and risk aversion (grains with long awns have high pest resistance according to farmers)–were also ranked high. We included the most prominent quality variable (*chapati* quality) in the choice experiment, being a most prominent trait in decision-making. After quality, farmers ranked traits indicative of higher yield (height of the plant, more

**Table 4. Initial ranking analysis on wheat varietal attributes.**

| | Overall (n = 120) | | Men (n = 51) | | Women (n = 69) | | SCST (n = 34) | | Other castes (n = 86) | |
|---|---|---|---|---|---|---|---|---|---|---|
| | Importance [Dummy] | Rank[#] | Importance [Dummy] | Rank[#] | Importance [Dummy] | Rank[#] | Importance [Dummy] | Rank[#] | Importance [Dummy] | Rank[#] |
| Good *chapati* quality | 0.936 | 2.39 (1.81) | 0.900 | 2.53 (1.65) | 0.960 | 2.31 (1.91) | 0.917 | 2.09 (1.65) | 0.944 | 2.51 (1.87) |
| Color of grain at maturity | 0.936 | 4.14 (1.67) | 0.980 | 4.80 (1.71) | 0.907* | 3.66*** (1.48) | 0.861 | 4.13 (1.71) | 0.966** | 4.14 (1.67) |
| Big and bold grains | 0.856 | 1.88 (0.94) | 0.830 | 2.09 (0.98) | 0.875 | 1.73** (0.88) | 0.944 | 1.76 (0.74) | 0.820* | 1.93 (1.02) |
| Length/spread of awns | 0.816 | 4.81 (2.14) | 0.920 | 5.26 (2.31) | 0.747** | 4.45* (1.93) | 0.944 | 4.59 (1.79) | 0.764** | 4.93 (2.30) |
| Height of plant | 0.760 | 5.16 (2.12) | 0.860 | 5.49 (2.56) | 0.693** | 4.88 (1.64) | 0.694 | 4.96 (1.40) | 0.787 | 5.23 (2.32) |
| Early maturity | 0.512 | 7.20 (2.44) | 0.680 | 7.32 (2.34) | 0.400*** | 7.07 (2.57) | 0.333 | 7.50 (1.68) | 0.584*** | 7.13 (2.59) |
| More tillers per plant | 0.472 | 4.68 (2.05) | 0.560 | 5.04 (2.32) | 0.413 | 4.35 (1.76) | 0.444 | 4.44 (2.19) | 0.483 | 4.77 (2.02) |
| Resistance to drought | 0.472 | 5.00 (1.98) | 0.520 | 5.50 (2.20) | 0.440 | 4.61* (1.73) | 0.472 | 6.00 (1.66) | 0.472 | 4.60*** (1.98) |
| Lodging tolerance | 0.432 | 5.93 (2.43) | 0.580 | 5.90 (2.60) | 0.333*** | 5.96 (2.28) | 0.500 | 4.78 (1.63) | 0.404 | 6.50*** (2.58) |
| Number of grains per tiller | 0.432 | 5.46 (1.92) | 0.440 | 5.86 (2.42) | 0.427 | 5.19 (1.47) | 0.444 | 5.06 (1.91) | 0.427 | 5.63 (1.92) |
| Resistance to diseases and pests | 0.312 | 5.97 (2.05) | 0.480 | 5.75 (2.19) | 0.200*** | 6.33 (1.80) | 0.194 | 6.14 (1.86) | 0.360* | 5.94 (2.11) |
| Late heat stress tolerance | 0.168 | 8.24 (2.07) | 0.240 | 9.33 (1.78) | 0.120* | 6.78*** (1.48) | 0.083 | 6.67 (2.08) | 0.202 | 8.50 (2.01) |

*Notes*: The Importance dummy indicates whether a respondent considered the given attribute important while selecting varieties. When presented with 12 traits (with an option to add more), women considered a lower number of traits as important than men, and this difference is statistically significant. Between the caste groups, no significant difference exists with respect to the average number of traits considered important.

[#]Ranking exercise was conducted among the traits considered important by the respondents.

Figures in parentheses show the standard deviation of the sample means. The inter-group (SCST vs. Other castes and Men vs. Women) equality of proportions and means is tested using large-sample statistics (*prtest* for importance dummy and *ttest* for ranks, using Stata 16.0).

***, **, and * show that the difference with the male / SCST group is significant at 0.01, 0.05, and 0.10 levels, respectively.

tillers per plant) above traits showing risk reduction (tolerance to drought, lodging, and diseases). During both the ranking exercise and the piloting of choice experiments, many farmers opined that 'grain yield in general' was more important in the variety selection than the individual yield components like the number of tillers. Based on this information, a quantity trait was included in the choice experiment: 'grain yield per land unit.' We included two risk- ameliorating factors from the ranking exercise–lodging tolerance (a familiar, high ranked trait) and terminal heat tolerance (an unfamiliar, low ranked trait). Because heat tolerance is expected to play an increasingly important role in wheat breeding, we provided information priming on this trait before presenting farmers with the alternatives to choose.

The choice experiment estimates are presented in Table 5, which are derived from 7,362 choices from 818 respondents in the main survey. In general, respondents preferred the improved varieties shown on the choice card over the status quo, as indicated by the positive ASC value of the basic aggregate model (*Model 1*, Table 5). Also, as expected, farmers' willingness to choose an improved variety declined with the seed price. The respondents preferred all attributes included in the choice experiment, as evident from the positive and statistically significant coefficients. However, the estimated parameter coefficients are not directly interpretable. The standard deviation parameters are statistically significant for three attributes–yield, high *chapati* quality, and seed price–indicating the existence of preference heterogeneity. In order to examine the source of this heterogeneity, we estimated additional models, interacting traits with household attributes.

Significant gender- and caste-specific preference heterogeneity can be expected in the willingness to pay values (*Model 2* vs. *Model 3*; *Model 4* vs. *Model 5*). Women also faced a steeper demand function than men, as evidenced by the highly negative coefficient of seed price (-0.005 *vs.* men's -0.001), showing that women's WTP for new varieties dropped drastically with an increase in the seed price. Most attributes included in the choice experiment were preferred by both men and women and both marginalized castes and others, as evident from the positive and statistically significant coefficients. The exceptions in the mean function were the high heat tolerance attribute in the women group and seed price in the SCST group. Both were insignificant. Women did not prioritize their decision based on high heat tolerance, and the non-marginalized households faced a steeper demand function. The standard deviation function indicates that female respondents expressed a significantly greater variability in preference for medium *chapati* quality than male respondents, whereas the opposite holds for high *chapati* quality. Male respondents expressed a large variability in preference for potential yield and medium lodging traits. Non-marginalized caste farmers showed greater variability than SCST farmers in preference for medium lodging.

Since the coefficients of the mixed logit model cannot be directly interpreted, we focus on the WTP values after including respondent-specific variables and curtailing the effect of ANA. The extent of ANA is shown in Fig 2. About 28% of respondents did not ignore any attributes presented as a decision simplification strategy. About 5% ignored all non-cost attributes. Lodging was the attribute ignored by most respondents during the decision-making (48%), while yield and quality were ignored by only a few (about 5% each). Heat tolerance was also not a factor ignored in the decision-making (only 9% ignoring), possibly due to the information priming. We addressed the ANA problem by excluding the 5% of respondents who ignored all non-cost attributes and re-estimated the WTP values, which are shown in *Model 7* (Table 6). The results show that the effect of excluding this minority, and therefore the effect of ANA, is highly pronounced in the WTP estimates of all traits included in the choice experiment. The ANA varied significantly between male and female respondents. Women ignored lodging more than men. While the ANA of the SCST and other castes was comparable in magnitude, the former paid more attention to the yield attribute.

**Table 5. Mixed logit model estimates on farmer preferences for varietal attributes.**

| | | Overall *Model 1.* | Gender groups | | Caste groups | |
|---|---|---|---|---|---|---|
| | | | *Model 2.* Female respondents | *Model 3.* Male respondents | *Model 4.* SCST households | *Model 5.*Other caste households |
| Heat tolerance [reference level: Low] | | | | | | |
| High tolerance | mean function | 0.335*** (0.056) | 0.118 (0.075) | 0.926*** (0.107) | 0.615*** (0.099) | 0.479*** (0.069) |
| | std. dev. function | 0.259 (0.396) | 0.345 (0.376) | 0.046 (0.391) | 0.078 (0.483) | 0.021 (0.372) |
| Medium tolerance | mean function | 0.587*** (0.069) | 0.464*** (0.091) | 0.900*** (0.126) | 0.807*** (0.120) | 0.642*** (0.084) |
| | std. dev. function | 0.083 (0.175) | 0.173 (0.155) | 0.051 (0.161) | 0.037 (0.178) | 0.083 (0.170) |
| Potential yield | mean function | 0.314*** (0.007) | 0.326*** (0.010) | 0.359*** (0.013) | 0.292*** (0.012) | 0.324*** (0.009) |
| | std. dev. function | 0.051*** (0.008) | 0.070*** (0.011) | 0.258*** (0.015) | 0.070*** (0.011) | 0.065*** (0.011) |
| *Chapati* quality [reference level: Low] | | | | | | |
| High | mean function | 2.350*** (0.098) | 2.461*** (0.128) | 2.464*** (0.167) | 1.959*** (0.171) | 2.565*** (0.121) |
| | std. dev. function | 0.782*** (0.202) | 0.032 (0.319) | 1.463*** (0.285) | 0.102 (0.315) | 0.383 (0.345) |
| Medium | mean function | 1.477*** (0.094) | 1.505*** (0.120) | 1.458*** (0.160) | 1.219*** (0.164) | 1.704*** (0.115) |
| | std. dev. function | 0.086 (0.285) | 1.078*** (0.217) | 0.093 (0.354) | 0.516* (0.279) | 0.469 (0.289) |
| Lodging tolerance [reference level: Low] | | | | | | |
| High | mean function | 1.694*** (0.055) | 1.666*** (0.076) | 2.050*** (0.105) | 1.613*** (0.094) | 1.586*** (0.068) |
| | std. dev. function | 0.263 (0.417) | 1.734*** (0.285) | 1.387*** (0.372) | -0.128 (0.441) | -0.128 (0.446) |
| Medium | mean function | 1.109*** (0.044) | 0.678*** (0.057) | 1.993*** (0.091) | 0.943*** (0.073) | 1.126*** (0.055) |
| | std. dev. function | 0.141 (0.127) | 0.583*** (0.175) | 2.652*** (0.188) | -0.068 (0.178) | 0.979*** (0.138) |
| Seed price | mean function | -0.003*** (2.E-04) | -0.005***(3.E-04) | -0.001***(3.E-04) | -2.E-04 (3.E-04) | -0.003*** (3.E-04) |
| | std. dev. function | 0.014*** (0.001) | 0.019*** (0.001) | 0.005*** (4.E-04) | 0.042*** (0.003) | 0.015*** (0.001) |
| ASC | | 1.531*** (0.291) | -2.486*** (0.407) | 5.904*** (0.415) | 4.106*** (0.539) | 1.386*** (0.341) |
| Number of observations | | 7,362 | 3,717 | 3,645 | 2,412 | 4,950 |
| Log Likelihood | | -6,611.53 | -3,549.85 | -2,711.70 | -2,408.04 | -4,422.48 |

*Note*: Coefficients are shown with standard errors in parentheses.

***, **, and * denote that the coefficients are statistically significant at 0.01, 0.05, and 0.10 levels, respectively.

We included the respondent-specific variables, re-estimated the model, and calculated the WTP (Table 6). Similar to *Model 1*, the coefficients of all traits included in the choice experiment were significant in both *Model 6* and *7*. Once the extreme ANA respondents were excluded, the ASC value increased in magnitude, indicating an increase in the WTP values in *Model 7*. The standard deviation parameter remained significant only for seed price and potential yield. Most of the respondent characteristics were statistically significant in both models. Due to word limits, we restrict the interpretation to the WTP values from *Model 6*.

An inter-trait comparison of the WTP values shows that farmers were willing to pay Rs. 615 per acre, which is about 7.8 times the WTP for an improved variety with an incremental yield potential of one quintal per acre. In other words, the value of grain quality is equivalent to an additional 7.8 quintals yield per acre (1.9 tons/ha). The WTP for moderate *chapati* quality was

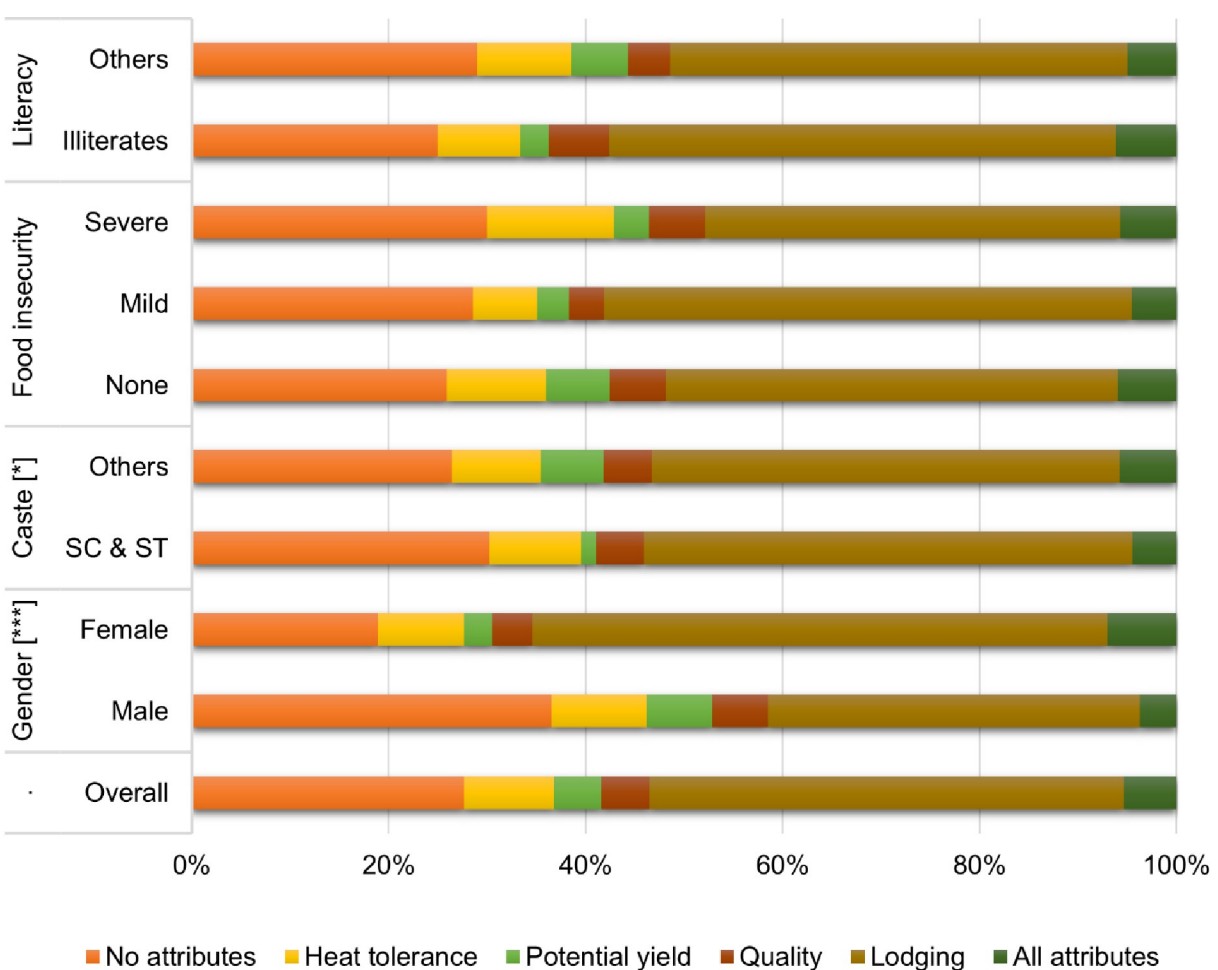

**Fig 2. Attribute non-attendance among respondent groups.** Note: *** and * represent statistically significant difference in the distribution of categories of attribute non-attendance between the sub-groups at 0.01 and 0.10 levels, respectively.

also high (equivalent to a yield increase of 1.2 tons/ha). Consistent with the ranking analysis, farmers were willing to pay a higher price for a familiar and observable risk–lodging–than for a less familiar risk–heat tolerance–that was relatively difficult for them to comprehend. The WTP value was Rs. 432/acre (US$ 15.2/ha) for varieties with high tolerance to lodging, but only Rs. 87/acre (US$ 3.1/ha) for varieties with high heat tolerance. Thus, wheat varieties with medium to superior *chapati* quality, medium to high lodging tolerance, and medium heat tolerance are more likely to be accepted by most smallholders than one with high yield potential under favorable production conditions *alone*. Similar patterns were obtained when the extreme ANA was addressed, although the magnitude of WTP increased for all the traits in *Model 7*.

## 4.3. Preference heterogeneity

**Gender-based heterogeneity of preferences.** The ranking analysis provided the first hint that female respondents had different trait preferences than men (Table 4). Women considered a smaller set of attributes (on average 6.5 per respondent) as important for varietal selection than men (on average 8.7 attributes). For example, grain color at maturity was an

**Table 6. Choice experiment models accounting for preference heterogeneity and attribute non-attendance.**

| | | Coefficients | | WTP in Rs. derived from | |
| --- | --- | --- | --- | --- | --- |
| | | Model 6.Overall | Model 7. Excluding those who ignored all attributes | Model 6. | Model 7. |
| Heat tolerance [reference level: Low] | | | | | |
| High tolerance | mean function | 0.371*** (0.056) | 0.384*** (0.061) | 86.97*** (14.46) | 129.10*** (23.76) |
| | std. dev. function | 0.087 (0.412) | 0.102 (0.674) | | |
| Medium tolerance | mean function | 0.590*** (0.070) | 0.656*** (0.077) | 138.36*** (18.28) | 220.83*** (31.25) |
| | std. dev. function | 0.164 (0.137) | 0.114 (0.196) | | |
| Potential yield | mean function | 0.337*** (0.007) | 0.362*** (0.008) | 78.91*** (3.32) | 121.74*** (6.89) |
| | std. dev. function | 0.035*** (0.010) | 0.054*** (0.011) | | |
| *Chapati* quality [reference level: Low] | | | | | |
| High | mean function | 2.622*** (0.100) | 2.856*** (0.115) | 614.68*** (34.08) | 961.35*** (65.51) |
| | std. dev. function | 1.510*** (0.152) | 0.049 (0.384) | | |
| Medium | mean function | 1.670*** (0.095) | 1.868*** (0.110) | 391.54*** (28.08) | 628.74*** (51.92) |
| | std. dev. function | 0.008 (0.273) | 0.433 (0.300) | | |
| Lodging tolerance [reference level: Low] | | | | | |
| High | mean function | 1.842*** (0.056) | 1.894*** (0.059) | 431.89*** (19.78) | 637.67*** (35.99) |
| | std. dev. function | 0.789*** (0.293) | 0.252 (0.509) | | |
| Medium | mean function | 1.236*** (0.045) | 1.234*** (0.047) | 289.78*** (14.05) | 415.41*** (24.29) |
| | std. dev. function | 0.129 (0.141) | 0.069 (0.192) | | |
| Seed price | mean function | -0.004*** (0.0002) | -0.003*** (2.E-4) | | |
| | std. dev. function | 0.011*** (4.E-4) | 0.006*** (3.E-4) | | |
| ASC | | 0.701 (0.435) | 3.479*** (0.457) | 164.37 (105.31) | 1,171.14*** (191.09) |
| *ASC interaction with* | | | | | |
| SCST | | 1.166*** (0.240) | 0.920*** (0.243) | 273.42*** (56.81) | 309.69*** (83.22) |
| Gender | | 0.592*** (0.220) | 0.713*** (0.220) | 138.74*** (51.76) | 240.13*** (74.93) |
| Gender x SCST | | -0.235 (0.389) | 0.088 (0.389) | -55.15 (91.37) | -29.70 (130.93) |
| Food Insecurity | | 0.184*** (0.051) | 0.149*** (0.051) | 43.07*** (12.09) | 50.24*** (17.15) |
| Gender x Food Insecurity | | 0.452*** (0.101) | 0.200** (0.096) | 105.87*** (23.82) | 67.30** (32.56) |
| Landholding | | -0.061* (0.034) | -0.108*** (0.037) | -14.36* (8.01) | -36.51*** (12.45) |
| Age | | -0.008 (0.007) | 0.004 (0.007) | -1.85 (1.59) | 1.50 (2.35) |
| Illiteracy | | 1.895*** (0.241) | 0.992*** (0.230) | 444.34*** (57.02) | 334.01*** (77.93) |
| Number of observations | | 7,362 | 6,966 | | |
| Log Likelihood | | -6,622.08 | -5,864.309 | | |

*Notes*:

***, **, and * denote that the coefficients are statistically significant at 0.01, 0.05, and 0.10 levels, respectively. Except for potential yield, the WTP values are estimated as Indian Rupees per acre. For potential yield, it is Indian Rupees per quintal of grain. The household-specific variables are defined in the methodology section.

important attribute for varietal selection for 98% of men but only 91% of women. Since only those traits considered important by the respondents are included subsequently in the ranking exercise, women's ranks were generally high. In the example of grain color trait, women's ranking (3.7) was significantly higher than men's (4.8). Due to this reason, the share of respondents considering a trait as important in the varietal selection would be more informative. Only a small share of women, as compared to men, considered lodging tolerance (33% vs. 58%) and heat tolerance (12% vs. 24%) as the important attributes in the ranking exercise with no budget constraint.

A similar observation can be derived from the alternative specific constant (ASC) value from the choice experiment model (Table 5; *Models 2 & 3*), which captures the effects on the

utility of any attributes not included in the choice experiment. The ASC is negative for women (Table 5), which signals women's general reluctance to opt for new technologies in wheat culti-vation. We found no significant differences between male and female respondents for *chapati* quality and potential yield. However, men were more likely to adopt varieties tolerant to heat stress and lodging than women. This observation is consistent with the ranking exercise, which is not constrained by the household income, in which women provided the risk- ame-liorating traits with a lower rank than men.

The positive ASC-Female interaction term suggests that, *among those who did not opt for the status quo option*, women's preference and WTP for new varieties were higher than their male counterparts. Female respondents had a lower ANA in the choice experiment (Fig 2), and after addressing this (excluding those with extreme ANA), the ASC-Gender interaction coeffi-cient increased significantly. The ASC-Gender-Food Insecurity interaction was statistically sig-nificant and positive, showing that economically disadvantaged women are more eager to adopt new varieties than the better-off women. On the other hand, the ASC-SCST-Gender interaction was insignificant, indicating no caste-gender intersectionality in varietal preferences.

**Caste-based heterogeneity of preferences.** The inter-household preference heterogeneity along caste lines was evident even in the initial ranking analysis (Table 4). There was a pro-nounced difference between SCST and other castes in whether or not a trait was considered important while choosing varieties. In relative terms, a lower share of SCST households (86%) than of non-SCST households (97%) considered grain color an important trait. A higher share of non-SCST households considered early maturity and disease resistance as important traits. On the other hand, big, bold grains and long awns were preferred by a larger share of SCST households (94% for both attributes) than of non-SCST households (82% and 76%, respec-tively). With respect to the attribute ranks, the only resistance to drought and lodging toler-ance showed significant inter-caste differences. Lodging tolerance was ranked high by SCST households, while resistance to pests was ranked high by other castes.

In the choice experiment models (Table 5), SCST households were found to have a higher ASC value, indicating a general openness towards the new technologies. The coefficients of attribute variables in the choice experiment were comparable between the SCST and the other caste groups. The mean WTP for the different attributes could be expected to be similar across the caste groups, as shown by the insignificant ASC-SCST interaction term (*Model* 6, Table 6). The SCST respondents had lower ANA (Fig 2), and after addressing this in *Model 7*, the ASC interaction with the SCST dummy became statistically significant (Table 6).

In Table 6, the ASC interactions with farmer illiteracy, age, landholding size, and food inse-curity score (0–8 in increasing severity of food insecurity as felt by the respondent) are also included. The illiteracy interaction yielded a positive coefficient and WTP, the magnitude of which did not change drastically after adjusting for extreme ANA in the sample. The WTP increased with an increase in the food insecurity scale, while the large farmers had a lower WTP than their smaller counterparts. In other words, socially and economically marginalized respondents were more pro toward adopting the new varieties. Having said that, their actual willingness to pay may be constrained by budget limitations and the bottlenecks of the local seed networks.

## 5. Discussion and implications

Madhya Pradesh has long remained in the second position with respect to wheat acreage in India [76]. As shown in Section 3.1, the wheat productivity in the state is lower than the national average, one of the reasons for which could be the suboptimal varietal turnover rate. We found that the average age of varieties cultivated by sample farmers was 30 years, which is

extremely high (and hence undesirable) as compared to the average varietal age of wheat in India (10–12 years) and the age recommended to curtail the risks associated with wheat rust (<6 years) [23]. This low turnover makes the system highly vulnerable to biotic stresses. Madhya Pradesh farmers cultivate several wheat varieties known for premium quality, with lustrous grains with a greater gluten strength [77]. However, only a small share (8 out of 34 notified varieties during 1965–2018) are popularized in this region as having superior *chapati* quality [78]. Possibly due to the weak extension and seed supply networks in rural India, only a few of the recently released varieties reach farmers' fields on time. In this section, we discuss some of the less-explored but critical elements of farmer preferences for wheat varieties for strategizing breeding and varietal dissemination activities.

## 5.1. Importance of consumption utility

We have observed that wheat farmers have a high preference for varieties suitable for making superior quality *chapatis*, characterized by full puffing, light brown color, and soft and pliable texture. We found that the wheat varieties that produce grains for quality *chapatis* were preferred, with a WTP equivalent to 1.2–1.9 tons/ha increase in productivity (*cf. Model 6*, Table 6). With the median wheat yield of sample farmers at 2.0 tons/ha, this high WTP emphasizes the importance of consumption utility in varietal selection. Its relevance is not unknown to the scientific community. For instance, C-306, a wheat variety released in 1965, is documented as popular among farmers in some parts of India due to its superior grain quality [78, 79]. One of the popular varieties in the study area, HI-617, is a selection from C-306. Such patterns indicate that, unless they meet farmers' quality expectations, the new, improved wheat varieties may not become popular in Central India.

Consumer demand for quality grain may translate to higher market prices, as observed in the case of vegetable crops [80]. However, the role of output markets in varietal choice was found to be limited in the present study. According to our household survey data, only half of the sample households sold their produce in the market. About 17% were able to sell part of their produce in the government markets (*mandis*) and obtain a higher price (the Minimum Support Price, or MSP, as set by the government) than in the private market. The price for wheat grain in the *mandis* is fixed by the government before every cropping season as an institutional mechanism for incentivizing farmers to adopt new technologies. The MSP is uniform for a crop across markets and varieties [81]. We did not find Lok-1 or *Sujata* growers getting a premium price from the private markets either. This observation is supported by Gandhi and Koshy [82]. The reasons for quality attributes not being reflected in the grain markets merit separate research. We can only conclude that since most farmers cultivate varieties suitable for self-consumption, and none get a price premium above the MSP for quality, the utility from subsistence consumption is a predominant determinant for the low varietal turnover in the study region. However, one should not underestimate the role of seed value chains and information networks. Most farmers depend on farm-saved seeds or seeds obtained from informal sources, in spite of their willingness to experiment with new varieties, as revealed in the choice experiment. We recommend (a) the development of value chains for wheat grains that are responsive to grain quality by providing price premiums and (b) the dissemination of seeds of new, high-yielding wheat varieties with premium quality grain as the two crucial market interventions that could increase farmers' income and utility.

## 5.2. Low preference for risk-ameliorating attributes

Farmers' indifference towards risk-ameliorating attributes was evident from the ranking exercise, the choice experiment estimates, as well as the set of varieties actually cultivated by them.

Tolerance to terminal heat was shown as the least essential attribute and ranked as the least preferred attribute across different gender and caste groups (Table 4). Assuming that one reason for this was the lack of information about the possible effects of terminal heat, we provided a description before conducting the choice experiment. Despite the information priming, farmers' WTP for heat tolerance was relatively low compared to their WTP for other attributes (Table 6). Most varieties grown on farmers' fields were not particularly tolerant to biotic and abiotic stresses. We have seen that women are more indifferent than men to various risk-ameliorating attributes, and this highlights the need for gender-targeted information campaigns to ensure the widespread adoption of stress-resistant wheat varieties in Central India.

More research is required to understand farmers' relative indifference towards the heat-tolerant attribute because of two reasons. One, wheat production is severely affected by warming atmospheric temperatures, particularly in central and eastern India [83]. Two, there is increasing investment in research-and-development programs for breeding and popularizing heat-tolerant wheat varieties in the Indian subcontinent [84].

### 5.3. Gender- and caste-based heterogeneities

From the preliminary ranking exercise itself, it was evident that both men and women valued quality attributes the most, quantity (yield) attributes the next, and risk-ameliorating attributes the least. Men and women put equally high emphasis on consumption utility. These patterns are also reflected in the choice experiment estimates. The coefficients of *chapati* quality attribute in the choice experiment are more or less equal in magnitude across models with male and female respondents (Table 5). Due to a shallow demand curve originating from higher economic freedom and agency in varietal selection, men's preferences might get translated to a high WTP value for *chapati* quality. This finding has significant academic relevance, as it contradicts the widely held belief that men only aim to increase the marketed surplus, whereas women, being custodians of family diets and nutrition, prioritize food security and prefer varieties that are both palatable and nutritious [85–87].

The caste-based preference heterogeneities have a different set of implications for the diffusion of varietal innovations. The mean WTP of SCST households was higher than the non-marginalized castes for new varieties in the hypothetical choice experiment (Table 6). In real life, however, these farmers cultivate older varieties (varietal age being 22% higher than other castes; Table 2). The disparity between the expressed willingness to adopt and the actual adoption of new varieties could be mainly due to bottlenecks in the seed supply chains and information networks, as well as households' inability to pay in monetary terms. The sub-optimal rate of access by farmers belonging to the marginalized castes to information, technologies, and markets is well-documented in the literature [49, 88, 89]. However, we provide one of the first strands of empirical evidence for the high willingness of farmers from the marginalized castes to move away from the status quo by adopting new promising varieties.

### 6. Conclusion

The present study identified the key characteristics of wheat varieties that smallholders preferred and examined the pattern of preference heterogeneity in Central India. Due to the high prevalence of income poverty and low wheat productivity in the study region, we had expected sample farmers to prioritize wheat varieties based on their productivity potential. However, grain quality was consistently ranked above yield-enhancing and risk-ameliorating factors. In fact, the prominence of quality traits could be a reason behind the prevalence of old varieties on-farm, resulting in low yield. This finding has grave implications for the research-and-development programs that aim to address the emerging challenges of climate change and biotic

stresses through breeding. In order to ensure wider farmer acceptability of climate-resilient and pest-resistant varieties within a reduced timespan, the new varieties should meet the basic minimum quality attributes (e.g., high molecular-weight protein content and Glu-1 score that are associated with *chapati* texture [90]), both in the subsistence-oriented production systems as well as in quality-conscious grain/flour markets.

The study has detected significant preference heterogeneity for varietal traits. Farmers belonging to the marginalized castes, women, illiterate farmers, and those who faced food insecurity were more willing to opt for new varieties than their counterparts in the hypothetical scenario of the choice experiment. More empirical research is needed to understand the pathways through which the preference heterogeneity within the farming community determines the rate of varietal turnover.

Finally, there are certain limitations to generalizing these findings to apply to a wider farmer population of India. The sample is limited to three districts of Central Madhya Pradesh, where the wheat production scenario is unique for the predominance of consumption utility in varietal selection. Farmers in other states such as Punjab and Haryana, India's breadbasket with well-integrated markets, have a relatively rapid varietal replacement and possibly different preferences for wheat varietal traits. Therefore, caution may be exerted while generalizing the findings. Furthermore, there could be variations in the nature and implications of gender- and caste-based preference heterogeneities across Indian states. While recommending a more participatory, farmer-oriented approach to developing the crop breeding programs, we also recognize the need for more socioeconomic research studies to be conducted across the wheat systems in India. Only through these studies, we can help identify wheat varieties that meet farmers' less explored but important preferences (e.g., taste, texture, etc.) and thereby increase the varietal turnover rate in rural India, particularly among marginalized social groups.

## Supporting information

**S1 Appendix. Example choice card.**
(DOCX)

**S2 Appendix. The extent of involvement of male and female members in wheat varietal selection.**
(DOCX)

**S1 Data.**
(CSV)

**S2 Data.**
(CSV)

**S1 File.**
(PDF)

## Acknowledgments

We are grateful for valuable comments and suggestions from two anonymous reviewers. We also thank Mr. Sreejith Aravindakshan (Arunachal University of Studies, Namsai) for his input in designing the choice experiment, Dr. Pankaj Singh (Borlaug Institute for South Asia, BISA, Jabalpur) for facilitating the fieldwork, and Mr. Kapil Kale and the rest of the *Survey Jena* team for monitoring and organizing the data collection activities.

## Author Contributions

**Conceptualization:** Vijesh V. Krishna.

**Data curation:** Vijesh V. Krishna.

**Formal analysis:** Vijesh V. Krishna, Prakashan C. Veettil.

**Funding acquisition:** Vijesh V. Krishna.

**Investigation:** Vijesh V. Krishna.

**Methodology:** Vijesh V. Krishna, Prakashan C. Veettil.

**Project administration:** Vijesh V. Krishna.

**Resources:** Vijesh V. Krishna.

**Supervision:** Vijesh V. Krishna.

**Validation:** Vijesh V. Krishna.

**Writing – original draft:** Vijesh V. Krishna, Prakashan C. Veettil.

**Writing – review & editing:** Vijesh V. Krishna.

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
