## [Decision Letter · Decision Letter 0]

16 Jul 2021

PONE-D-21-06519

Gender, Caste, and Farmer Preference for Wheat Varietal Traits in Rural India

PLOS ONE

Dear Dr. Krishna,

Thank you for submitting your manuscript to PLOS ONE. After careful consideration, we feel that it has merit but does not fully meet PLOS ONE’s publication criteria as it currently stands. Therefore, we invite you to submit a revised version of the manuscript that addresses the points raised during the review process.

We look forward to receiving your revised manuscript.

Kind regards,

Rosalyn B. Angeles-Shim, PhD

Academic Editor

PLOS ONE

Journal Requirements:

Reviewers' comments:

Reviewer's Responses to Questions

**Comments to the Author**

1. Is the manuscript technically sound, and do the data support the conclusions?

Reviewer #1: Partly

Reviewer #2: Partly

2. Has the statistical analysis been performed appropriately and rigorously? 

Reviewer #1: Yes

Reviewer #2: No

3. Have the authors made all data underlying the findings in their manuscript fully available?

Reviewer #1: No

Reviewer #2: Yes

4. Is the manuscript presented in an intelligible fashion and written in standard English?

Reviewer #1: Yes

Reviewer #2: No

5. Review Comments to the Author

Reviewer #1: The paper sets out to assess the preferences of central Indian farmers towards wheat varieties with a special attention to differences among gender and castes. Following an initial ranking exercise, a choice experiment involving five traits (heat tolerance, chapati quality, lodging, seed price and yield) is conducted; results are analysed through a mixed logit model correcting for attribute non-attention).

The paper makes some valuable contributions to studying farmers' preferences for varietal traits. Although the main analysis seems sounds, the presentation is not optimal leading to confusion. There are also some claims in the discussion that seem either unsupported by the data or vague/disconnected. I made several suggestions to clarify and sharpen the discussion but none of these are insurmountable. I highlighted the major comments in yellow.

See more details in the review report attached

Reviewer #2: The different parts of the manuscript are mixed. There are some results in the methodology section and some discussions in the results sections. Authors did first a ranking exercise followed by a household survey. However, the link between the two parts is not clear. How the ranking has informed the experimental choice exercise need to be clarified.

6. PLOS authors have the option to publish the peer review history of their article (what does this mean?). If published, this will include your full peer review and any attached files.

Reviewer #1: No

Reviewer #2: No

---

## [Author Response · Author response to Decision Letter 0]

2 Nov 2021

We are grateful to the editor and the two reviewers for their comprehensive reports and numerous suggestions to increase the value of the manuscript. While having a critical look at our manuscript, both of them found the topic valuable and supported the approach. We have carefully addressed each of the comments provided by the two reviewers. We feel that the paper is now much stronger.

---

## [Decision Letter · Decision Letter 1]

13 Apr 2022

PONE-D-21-06519R1Gender, Caste, and Heterogeneous Farmer Preferences for Wheat Varietal Traits in Rural IndiaPLOS ONE

Dear Dr. Krishna,

Thank you for submitting your manuscript to PLOS ONE. After careful consideration, we feel that it has merit but does not fully meet PLOS ONE’s publication criteria as it currently stands. Therefore, we invite you to submit a revised version of the manuscript that addresses the points raised during the review process.

We look forward to receiving your revised manuscript.

Kind regards,

Rosalyn B. Angeles-Shim, PhD

Academic Editor

PLOS ONE

Journal Requirements:

Reviewers' comments:

Reviewer's Responses to Questions

**Comments to the Author**

1. If the authors have adequately addressed your comments raised in a previous round of review and you feel that this manuscript is now acceptable for publication, you may indicate that here to bypass the “Comments to the Author” section, enter your conflict of interest statement in the “Confidential to Editor” section, and submit your "Accept" recommendation.

Reviewer #1: (No Response)

2. Is the manuscript technically sound, and do the data support the conclusions?

Reviewer #1: Yes

3. Has the statistical analysis been performed appropriately and rigorously? 

Reviewer #1: Yes

4. Have the authors made all data underlying the findings in their manuscript fully available?

Reviewer #1: Yes

5. Is the manuscript presented in an intelligible fashion and written in standard English?

Reviewer #1: Yes

6. Review Comments to the Author

Reviewer #1: The authors clearly improved the manuscript and added additional information. However, some ambiguity remains and should be addressed before the paper can be considered for publication.

Section 3

• I am still confused by the data sampling process. You say you rely on a previously conducted CIMMYT study, but a number of things are missing:

o Reference to this study (even if only to a working paper)

o What kind of information is used from this CIMMYT study? You explain a sampling strategy but it is unclear whether you or the CIMMYT study has done this. You also seem to refer to households and the "2018 respondent list", but later on, you refer to "after completing a census of farm households": why is this census needed and when was it conducted.

o You also state that "we excluded the first-round dataset from the main analysis" but later on add that "We supplement Table 3 with data on gendered participation in wheat varietal selection, using the 2018 survey data". This seems contradictory.

• The sentence "The social norms that vary across the caste 35 groups could influence the gender relations in farm production." Is redundant.

• Eq. 1, 2, 5 and 6 seems not strictly necessary: the most important is to show your final formula to be estimated (Eq. 3-4) and how the WTP is calculated (ratio of trait coefficient over seed price coefficient). The use of too many Greek symbols is lowering the readability. Also, p. 16 (manuscript numbering) lines 9-10 seems to contradict line 18: are the parameters assumed random or not?

Section 4

• "In our survey, the male respondents also did not acknowledge the role of women members": what does this mean and does this apply to every single male respondent in the entire sample?

• Table 4: " inter-group equality": male vs female, SCST vs other castes?

• P 19 l 7-8: "There exists significant preference heterogeneity within the gender groups, as shown by the statistically significant standard deviation parameter for most of the traits.": this seems confusing for two reasons:

o I only see a difference in standard deviation for chapati quality, so the statement seems an overgeneralisation

o "preference heterogeneity within the gender groups" is also confusing; it might be more accurate to say something like: Female respondents expressed a significantly higher variability in preference for high Chapati quality than male respondents, and oppositely for medium Chapati quality.

• P19 l 17-19: "The results show that the effect of excluding this minority, and therefore the effect of ANA, is highly pronounced in the preference estimates of all traits included in the choice experiment.": the term highly pronounced seems exaggerated: only high Chapati quality, high lodging tolerance and ASC is significantly affected.

7. PLOS authors have the option to publish the peer review history of their article (what does this mean?). If published, this will include your full peer review and any attached files.

Reviewer #1: **Yes: **Bert Lenaerts

---

## [Author Response · Author response to Decision Letter 1]

19 May 2022

The response to Reviewer-1's comments are attached.

---

## [Decision Letter · Decision Letter 2]

14 Jul 2022

Gender, Caste, and Heterogeneous Farmer Preferences for Wheat Varietal Traits in Rural India

PONE-D-21-06519R2

Dear Dr. Krishna,

We’re pleased to inform you that your manuscript has been judged scientifically suitable for publication and will be formally accepted for publication once it meets all outstanding technical requirements.

Kind regards,

Rosalyn B. Angeles-Shim, PhD

Academic Editor

PLOS ONE

Additional Editor Comments (optional):

Reviewers' comments:

Reviewer's Responses to Questions

**Comments to the Author**

1. If the authors have adequately addressed your comments raised in a previous round of review and you feel that this manuscript is now acceptable for publication, you may indicate that here to bypass the “Comments to the Author” section, enter your conflict of interest statement in the “Confidential to Editor” section, and submit your "Accept" recommendation.

Reviewer #1: All comments have been addressed

2. Is the manuscript technically sound, and do the data support the conclusions?

Reviewer #1: Yes

3. Has the statistical analysis been performed appropriately and rigorously? 

Reviewer #1: Yes

4. Have the authors made all data underlying the findings in their manuscript fully available?

Reviewer #1: Yes

5. Is the manuscript presented in an intelligible fashion and written in standard English?

Reviewer #1: Yes

6. Review Comments to the Author

Reviewer #1: No more comments, everything seems resolved. I want to congratulate the authors for their continued effort in improving the paper.

7. PLOS authors have the option to publish the peer review history of their article (what does this mean?). If published, this will include your full peer review and any attached files.

Reviewer #1: **Yes: **Bert Lenaerts

---

## [Editor Report · Acceptance letter]

22 Jul 2022

PONE-D-21-06519R2 

Gender, Caste, and Heterogeneous Farmer Preferences for Wheat Varietal Traits in Rural India 

Dear Dr. Krishna:

I'm pleased to inform you that your manuscript has been deemed suitable for publication in PLOS ONE. Congratulations! Your manuscript is now with our production department. 

Kind regards, 

on behalf of

Dr. Rosalyn B. Angeles-Shim 

Academic Editor

PLOS ONE